# Symbiont-induced odorant binding proteins mediate insect host hematopoiesis

**Joshua B Benoit[1], Aurélien Vigneron[2], Nichole A Broderick[3,4], Yineng Wu[2], Jennifer S Sun[5], John R Carlson[5,6], Serap Aksoy[2], Brian L Weiss[2]\***

[1]Department of Biological Sciences, University of Cincinnati, Cincinnati, United States; [2]Department of Epidemiology of Microbial Diseases, Yale School of Public Health, New Haven, United States; [3]Department of Molecular and Cell Biology, University of Connecticut, Storrs, United States; [4]Institute for Systems Genomics, University of Connecticut, Storrs, United States; [5]Department of Molecular, Cellular and Developmental Biology, Yale University, New Haven, United States; [6]Interdepartmental Neuroscience Program, Yale University, New Haven, United States

**Abstract** Symbiotic bacteria assist in maintaining homeostasis of the animal immune system. However, the molecular mechanisms that underlie symbiont-mediated host immunity are largely unknown. Tsetse flies (*Glossina* spp.) house maternally transmitted symbionts that regulate the development and function of their host's immune system. Herein we demonstrate that the obligate mutualist, *Wigglesworthia*, up-regulates expression of *odorant binding protein six* in the gut of intrauterine tsetse larvae. This process is necessary and sufficient to induce systemic expression of the hematopoietic RUNX transcription factor *lozenge* and the subsequent production of crystal cells, which actuate the melanotic immune response in adult tsetse. Larval *Drosophila's* indigenous microbiota, which is acquired from the environment, regulates an orthologous hematopoietic pathway in their host. These findings provide insight into the molecular mechanisms that underlie enteric symbiont-stimulated systemic immune system development, and indicate that these processes are evolutionarily conserved despite the divergent nature of host-symbiont interactions in these model systems.

**\*For correspondence:** brian. weiss@yale.edu

**Competing interests:** The authors declare that no competing interests exist.

## Introduction

Mutualistic bacteria are functionally critical to the physiological well-being of their animal hosts. These microbes benefit their hosts by providing essential nutrients, aiding in digestion and maintaining intestinal equilibrium (*Douglas, 2015*; *Marchesi et al., 2016*). Additionally, mutualistic symbionts promote the development, differentiation and proper function of their host's immune system (*Chu and Mazmanian, 2013*; *Khosravi et al., 2014*; *Gomez de Agüero et al., 2016* ). Despite such an important role, the molecular mechanisms that underlie symbiont-mediated homeostasis of animal immunity remain largely unknown.

Insect models are useful for studying host-microbe interactions because, relative to their mammalian counterparts, they generally house taxonomically simple bacterial communities that can be easily manipulated during their host's development. Tsetse flies (*Glossina* spp.) house two gut-associated bacterial symbionts, obligate *Wigglesworthia glossinidia* and commensal *Sodalis glossinidius* (*Maltz et al., 2012*; *Wang et al., 2013*). In adult flies *Wigglesworthia* resides within cells that collectively form a bacteriome organ that is attached to the anterior midgut. *Sodalis* can be found

**eLife digest** Bacteria live within all animals. While a small number of these microbes can cause disease, most promote the health and wellbeing of their host. Microbes that support their host and benefit from the close association are often referred to as symbionts. Animals can be negatively affected and even become diseased if their symbionts are disrupted. As a result, a more complete understanding of the molecular interactions between animal hosts and their beneficial microbes will lead to better treatments for a number of diseases.

Tsetse flies are insects that harbor two bacterial symbionts, which are transferred from pregnant females to their larval offspring. If the offspring mature without these microbes, they fail to develop cells called hemocytes. These cells are normally found in the insect's equivalent of blood – a fluid called hemolymph – and they comprise an important component of the insect's immune system. Adult tsetse flies that lack hemocytes are susceptible to certain infections. These findings indicate that the bacterial symbionts induce the production of hemocytes in tsetse fly larvae via an unknown mechanism.

Benoit et al. now reveal that the bacterial symbionts trigger tsetse flies to produce a small protein called "odorant binding protein 6". This protein controls the generation of one specific type of hemocyte called crystal cells in developing larvae. Crystal cells are largely responsible for triggering the production of melanin, a protein involved in killing disease-causing microbes and inhibiting the loss of hemolymph from wound sites in the insect's exoskeleton.

Benoit et al. discovered that bacterial symbionts associated with the larvae of fruit flies also support the development of their host's immune system. Although these symbionts are acquired from the external environment rather than from the insect's parent, they too control the production of an odorant binding protein and crystal cells in their larval host.

Collectively, these findings confirm that bacterial symbionts are critically important for the development of the immune systems of insects, and they show that this process has been conserved throughout evolution. Future studies are likely to focus on identifying which molecules from the symbionts stimulate their hosts to produce new hemolymph cells. Furthermore, identifying which tissues and cell types in the animal hosts are targets for these molecules will provide a more complete picture of the pathways that lead to the production of new hemolymph cells.

extracellularly in the gut lumen, or intracellularly within gut epithelial cells (*Wang et al., 2013*). Tsetse reproduce via adenotrophic viviparity, during which pregnant females give birth to one larva each reproductive, or 'gonotrophic' (GC), cycle. Individual larvae mature through three developmental instars within the uterus, all the while receiving nourishment in the form of a milk-like substance produced by a modified accessory gland (milk gland; *Benoit et al., 2015*). Both *Wigglesworthia* and *Sodalis* are also found extracellularly in tsetse milk, and these bacteria colonize the gut of developing intrauterine larvae as they imbibe this nutrient source (*Attardo et al., 2008*).

Tsetse that undergo larvagenesis in the absence of their indigenous microbiota are highly immuno-compromised as adults (*Wang et al., 2013*). These 'aposymbiotic' (hereafter referred to as '*Gmm*^Apo') flies exhibit a dysfunctional cellular immune system that is characterized by the conspicuous absence of hemocytes. This phenotype results from the disruption of hematopoiesis (blood cell differentiation) during larval development (*Weiss et al., 2011*, *2012*). In insects, distinct hemocyte lineages mediate essential immune-related functions, including the phagocytosis and encapsulation of foreign invaders and the closing of cuticular wounds via the deposition of melanin at the site of injury (*Lemaitre and Hoffmann, 2007*; *Hillyer and Strand, 2014*; *Lee and Miura, 2014*). These immune mechanisms serve as the first line of defense following systemic challenge with exogenous organisms. Actuation of host immune system development represents an evolution driven mechanism that steadfastly links the tsetse fly with its symbiotic bacterial partners.

In this study we identify symbiont regulated genes and pathways in tsetse larvae using RNA-seq. Functional studies revealed that one symbiont induced gene, which encodes an odorant binding protein (OBP), regulates hematopoietic pathways during tsetse's larval development. We also demonstrate that *Drosophila*'s indigenous microbiota regulates expression of an orthologous,

functionally conserved OBP-encoding gene. Our findings detail a newly characterized, evolutionarily conserved component of a blood cell differentiation regulatory pathway that occurs in response to the presence of enteric symbionts.

## Results

### Enteric symbionts stimulate expression of *odorant binding protein 6* in intrauterine tsetse larvae

*Drosophila* hematopoiesis occurs primarily during early larval development (*Evans and Banerjee, 2003*). This process is likely similarly timed in wild-type tsetse, but fails to occur when larvae develop in the absence of their indigenous symbiotic bacteria (*Weiss et al., 2011*, *2012*). We sequenced RNA transcripts from age-matched (first and second instar) $Gmm^{WT}$ and $Gmm^{Apo}$ (generated as described in Materials and methods, *Fly lines and bacteria*) larvae in an effort to identify genes and pathways associated with hematopoiesis during tsetse larvagenesis. RNA-seq analysis revealed that 1166 genes exhibited a differential expression profile in $Gmm^{WT}$ compared to $Gmm^{Apo}$ larvae, and approximately 76% of these genes were expressed at higher levels in $Gmm^{WT}$ individuals (*Figure 1A*; *Supplementary file 1*). Gene ontogeny analysis revealed significant enrichment of genes functionally associated with B vitamin metabolism, larval development, organismal growth and chitin synthesis in the $Gmm^{WT}$ compared to $Gmm^{Apo}$ larvae (*Figure 1B*). These genes likely underlie previously observed phenotypes associated with dysfunctional chitin generation and B vitamin metabolism in $Gmm^{Apo}$ flies (*Weiss et al., 2013*; *Michalkova et al., 2014*). Herein we found that specific orthologues putatively clustering within the 'hematopoiesis' gene ontology category (GOC) were enriched in $Gmm^{WT}$ larvae compared to adult female and male flies (*Figure 1—figure supplement 1*). However, genes associated with hematopoiesis were not significantly enriched in $Gmm^{WT}$ compared to $Gmm^{Apo}$ larvae (*Figure 1—figure supplement 1*; *Supplementary file 2*). Because our previous studies demonstrate that $Gmm^{Apo}$ larvae fail to develop hemocytes (*Weiss et al., 2011*, *2012*), we hypothesized that factors not typically grouped within the hematopoiesis GOC likely induce hemocyte differentiation in $Gmm^{WT}$ larvae.

When our RNA-seq libraries were screened to identify highly abundant (TPM $\geq 10^3$) and differentially transcribed genes, we observed that tsetse *odorant binding protein 6* (*obp6*) exhibited the ninth highest level of differential expression between all annotated genes present in the larval $Gmm^{WT}$ and $Gmm^{Apo}$ libraries (*Supplementary file 1*). Specifically, $Gmm^{WT}$ larvae expressed 22x more *obp6* transcripts than did their $Gmm^{Apo}$ counterparts (*Figure 1C*; *Supplementary file 1*). Because chemosensory-related genes exhibit hematopoietic properties and immune system-associated expression profiles in other insects (*Thomas et al., 2016*; *Shim et al., 2013a*; *Aguilar et al., 2005*; *Bartholomay et al., 2004*; *Sabatier et al., 2003*), we investigated the functional relationship between *obp6* and immune system maturation processes during tsetse larvagenesis. *Obp6*, which encodes a 145 amino acid protein (16kD) with an N-terminal secretion signal (*Liu et al., 2010*), is larvae-enriched (*Figure 1D*) and the only OBP-encoding gene expressed at significantly different levels between $Gmm^{WT}$ and $Gmm^{Apo}$ individuals (*Figure 1E*; *Supplementary file 1*). *Obp6* expression can be restored in $Gmm^{Apo}$ larvae when their symbiont-cured moms are fed a diet supplemented specifically with *Wigglesworthia* cell extracts (*Figure 1F*), thus demonstrating that expression of this gene is stimulated by a *Wigglesworthia* derived molecule(s). Furthermore, this stimulus is likely not a bacterium generated nutrient, as the vitamin rich yeast extract included with the supplements fails to elicit the same response in individuals of the other treatment and control groups.

### RNAi-based trans-generational inhibition of *obp6* expression in tsetse larvae

$Gmm^{Apo}$ larvae express 22-fold fewer *obp6* transcripts than do their $Gmm^{WT}$ counterparts (*Supplementary file 1*), and $Gmm^{Apo}$ adults present a highly depleted population of hemocytes (*Weiss et al., 2012*). Equipped with this information, we set out to determine if *obp6* influences larval hemocyte differentiation processes and the subsequent function of these cells during adulthood. To do so we experimentally reduced *obp6* expression in intrauterine $Gmm^{WT}$ larvae using a novel RNAi-based trans-generational gene knock down approach (a graphical representation of the experimental design is presented in *Supplementary file 3*). To coincide with larval eclosion and

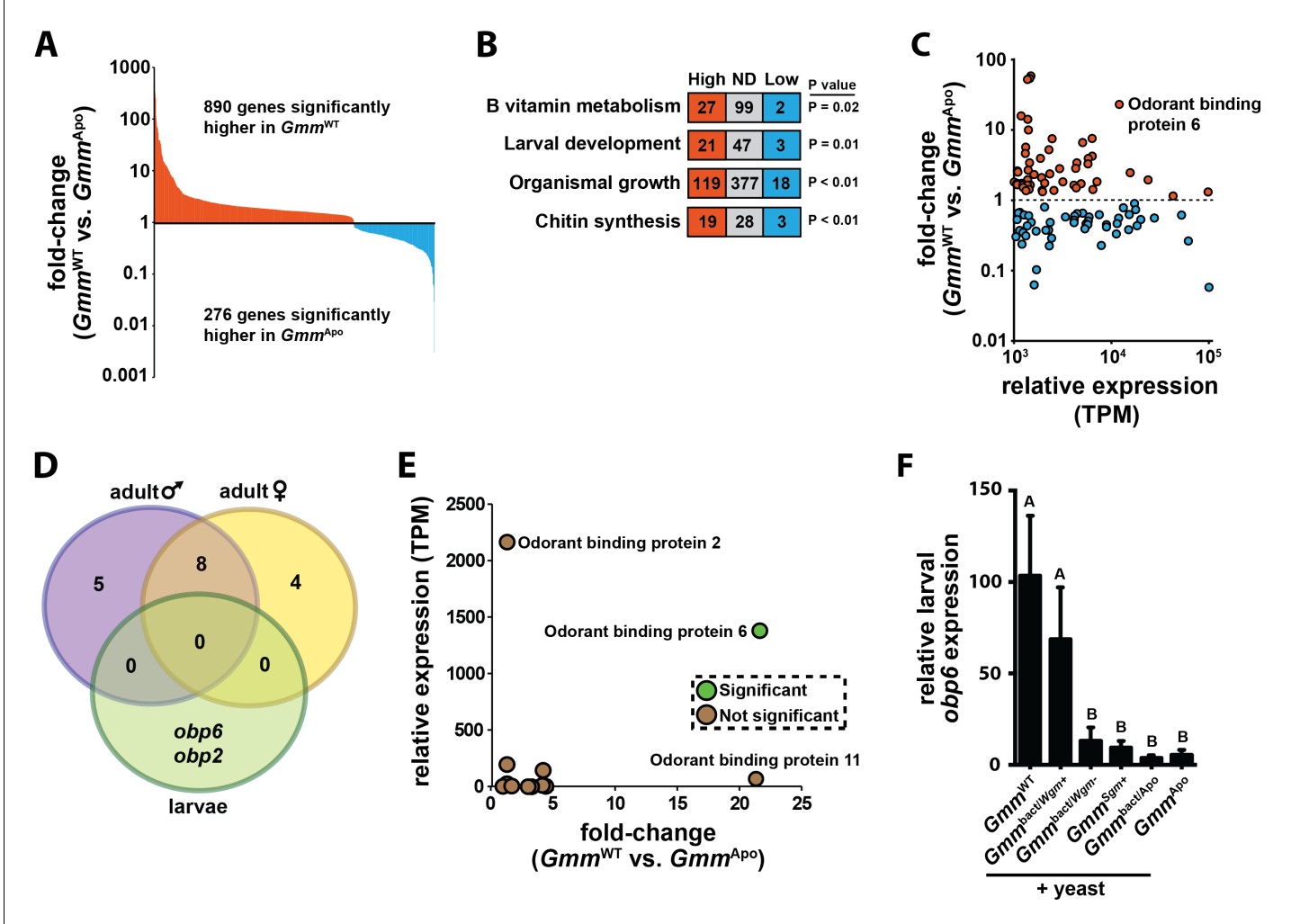

**Figure 1.** Symbiont-mediated differential expression of *odorant binding protein 6* in tsetse larvae. (A) Number of genes exhibiting significant differential expression, and a relative transcript abundance [in transcripts per million (TPM)] over 3, in *Gmm*WT compared to *Gmm*Apo larvae. Significance is based on a Baggerly's test followed by a false detection rate correction (p<0.01). (B) Significantly enriched gene ontology categories, determined using a Fisher's exact test. (C) Genes exhibiting significant differential expression (measure as fold-change in gene expression) in *Gmm*WT compared to *Gmm*Apo larvae, and a minimum TPM value of 1000. Significance was determined as in (A). (D) Enrichment analysis of odorant binding protein-encoding genes expressed in *Gmm*WT adult males (purple) and females (yellow), and *Gmm*WT larvae (green). (E) Relative expression (TPM) of tsetse odorant binding protein-encoding genes in *Gmm*WT larvae, and their differential expression (measure as fold-change in gene expression) in *Gmm*WT compared to *Gmm*Apo larvae. Significance was determined as in (A). (F) Relative *obp6* expression in *Gmm*WT larvae, as well as larvae derived from symbiont-cured moms fed a diet supplemented with yeast and *Wigglesworthia*-containing bacteriome extracts (*Gmm*bact/Wgm+), *Wigglesworthia*-free bacteriome extracts (*Gmm*bact/Wgm-), *Sodalis* cell extracts (*Gmm*Sgm+), and bacteriome extracts harvested from *Gmm*Apo females (*Gmm*bact/Apo). *Gmm*WT and *Gmm*Apo flies served as controls. n = 6 biological replicates for groups *Gmm*WT, *Gmm*bact/Wgm+ and *Gmm*Sgm+ samples, and n = 5 biological replicates for *Gmm*bact/Wgm-, *Gmm*bact/Apo and *Gmm*Apo samples. Replicates for all groups contain a mixture of four first and second instar larvae. Data are presented as mean ± SEM. Bars with different letters indicate a statistically significant difference (specific p values are listed in *Figure 1—source data 1*) between samples. Statistical analysis = ANOVA followed by Tukey's HSD post-hoc analysis.

The following source data and figure supplement are available for figure 1:

**Source data 1.** Obp6 expression in aposymbiotic tsetse larvae following supplementation.

**Figure supplement 1.** Developmental stage-specific enrichment analysis of tsetse orthologues that putatively cluster within the 'hematopoiesis' COG.

subsequent milk uptake, pregnant $Gmm^{WT}$ females were injected with either anti-$obp6$ (two anti-$obp6$ siRNAs were used, one of which was conjugated to a Cy3 dye) or anti-$gfp$ siRNAs on days 8 and 11 post-mating. siRNA-administered treatment (anti-$obp6$) and control (anti-$gfp$) moms, and their larval and adult offspring, are hereafter designated 'siOBP6' and 'siGFP', respectively.

Three days after the second treatment, pregnant females were viewed under a fluorescent microscope and siRNA was observed to have diffused throughout their hemocoel (*Supplementary file 4A*, top left panel). Additionally, siRNAs were taken up by the maternal milk gland and imbibed by developing (First gonotrophic cycle, GC1) intrauterine larvae, which subsequently fluoresced orange (*Supplementary file 4A*, bottom left panel). Obp6 expression was reduced in $Gmm^{WT}$ larvae by an average of 68% when they acquired corresponding siRNAs trans-generationally from their mother's milk (*Supplementary file 4B*). Finally, by the third gonotrophic cycle (26 days post-siRNA treatment), anti-$obp6$ siRNAs were no longer visible in treated moms or their offspring (*Supplementary file 4A*, top and bottom right panels, respectively), and larval $obp6$ expression had rebounded to levels equivalent to that found in GC1 control (siGFP) larvae (*Supplementary file 4B*). These recovered flies are hereafter designated 'siOBP6$^R$'.

## Obp6 expression in larval tsetse does not regulate the production or function of phagocytic hemocytes

The capacity of an adult insect to survive systemic challenge with exogenous microbes depends largely on the efficacy of its cellular immune system, and more specifically, hemocyte-mediated phagocytosis (*Hillyer and Strand, 2014*; *Vlisidou and Wood, 2015*). We thus investigated the ability of siOBP6, siGFP and siOBP6$^R$ adults to survive following systemic challenge with $10^3$ CFU of $E.$ $coli$ K12. We observed that 88% of siOBP6 adults, 12% siGFP adults and 4% of siOBP6$^R$ adults perished over the course of the experiment (*Figure 2A*). The lethal effect of $E.$ $coli$ K12 on siOBP6 adults, which is similar to that observed in $Gmm^{Apo}$ adults following exposure to the same challenge (*Weiss et al., 2012*), indicates that tsetse must express $obp6$ during larvagenesis in order for subsequent adults to survive following thoracic exposure to a needle-inoculated $E.$ $coli$ K12 challenge.

siOBP6 adults perish unusually fast following subjection to a hemocoelic challenge with $E.$ $coli$ K12. We thus investigated whether these flies house a depleted population of phagocytic hemocytes, which compromise the majority of the collective hemocyte population (*Kurucz et al., 2007*). We discovered no significant difference in the number of circulating hemocytes present in the hemocoel of siOBP6 compared to siGFP and siOBP6$^R$ adults (1349 ± 56, 1365 ± 33 and 1413 ± 31 hemocytes per µl of hemolymph, respectively; *Figure 2B*), and microscopic examination of hemolymph revealed that hemocytes from adult individuals of all three groups actively engulfed $E.$ $coli$ cells (*recE.* $coli_{GFP}$; *Figure 2C*). Finally, we observed that $E.$ $coli$ density initially increased during the first two days following injection into the hemocoel of siOBP6, siGFP and siOBP6$^R$ tsetse (3278 ± 806, 3530 ± 482 and 4085 ± 442 CFU per µl of hemolymph, respectively), but by four days later, had decreased to levels below that of the initial inoculate (104 ± 19, 115 ± 29 and 111 ± 28 CFU per µl of hemolymph, respectively; *Figure 2D*). These findings indicate that adult siOBP6 tsetse perish following systemic challenge with $E.$ $coli$ K12, but that this phenotype is not due to a reduced or dysfunctional population of phagocytic hemocytes. Additionally, $E.$ $coli$ growth does not appear to cause death in these flies, as bacterial density within both siOBP6 and siGFP (which survive this systemic challenge) individuals is maintained at similar densities during the course of infection.

## Larval tsetse must express obp6 in order for subsequent adults to produce melanin

We found that adult siOBP6 tsetse perish following systemic challenge with $E.$ $coli$ K12, but that this outcome surprisingly does not result from either a lack of functional phagocytic hemocytes or unimpeded bacterial replication. The insect cellular immune response also includes the synthesis of melanin, which is involved in the encapsulation of foreign organisms in the hemocoel as well as the deposition of melanin at the site of cuticular wounds (*Babcock et al., 2008*; *Tang, 2009*). We next investigated whether adult siOBP6 tsetse present a defective melanization cascade, and as such are unable to deposit melanin at the wound site inflicted during the systemic $E.$ $coli$ challenge procedure. We used a heat-sterilized glass needle to puncture the cuticle of siOBP6, siGFP and siOBP6$^R$ individuals, and monitored percent survival over time. Similar to their counterparts that were

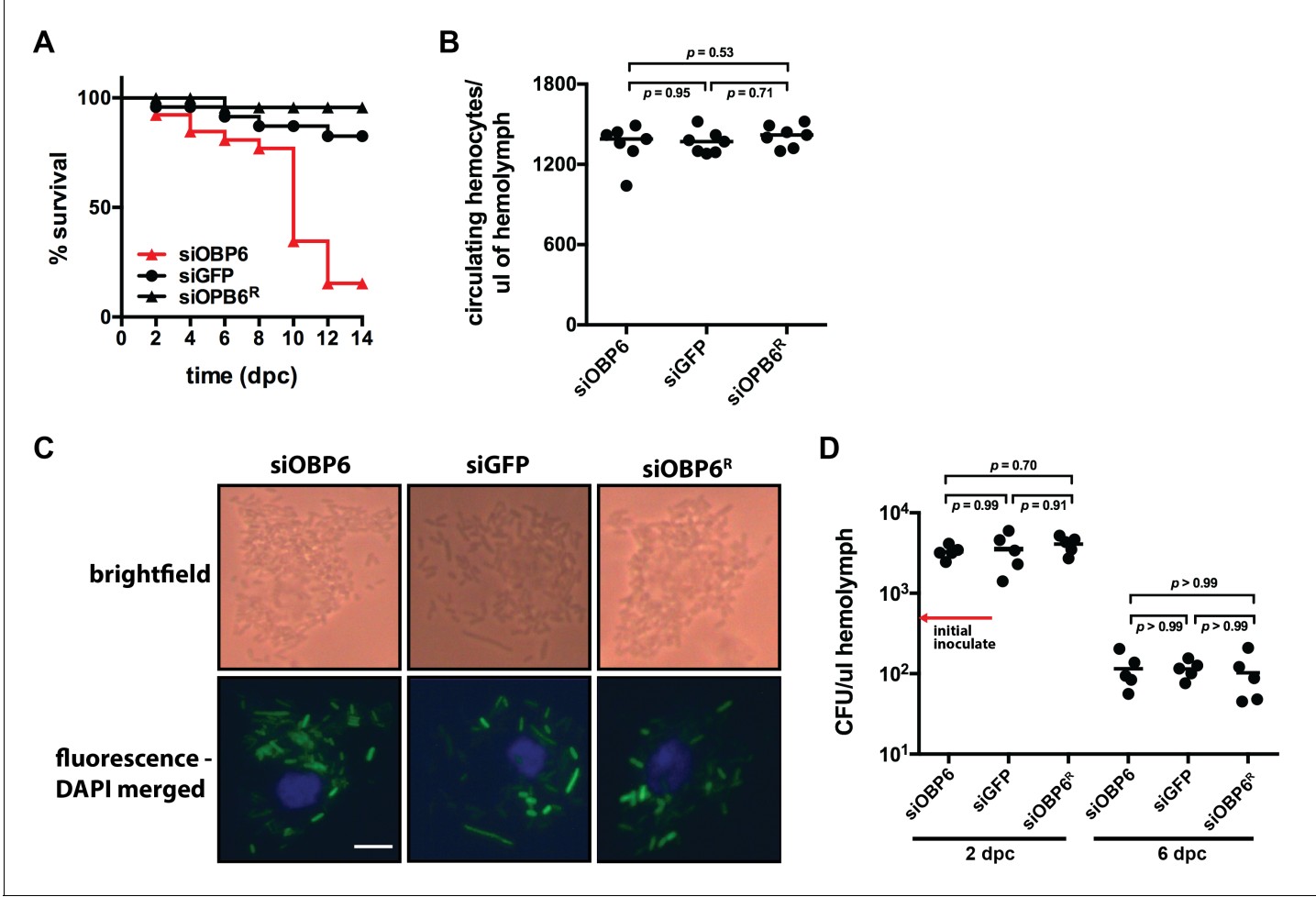

**Figure 2.** Tsetse *odorant binding protein 6* does not mediate the development and function of phagocytic hemocytes. (A) Survival following systemic challenge of siOBP6 and siGFP adults with $5 \times 10^2$ CFU of *E. coli* K12. Fly survival was monitored every other day for the duration of the 14 day experimental period. Survival assays were performed in triplicate, using 25 flies per replicate. Red curve depicts a statistically significant difference in infection outcome (p<0.0001, log-rank test). (B) Hemocyte abundance in siOBP6 and siGFP adults was quantified microscopically using a hemocytometer (*Figure 2—source data 1*). (C) A representative micrograph of hemocyte-engulfed *recE. coli*GFP from siOBP6, siGFP and siOBP6R adults. Experiment was performed using hemolymph collected from four distinct flies per (*Figure 2—source data 2*). Hemolymph was collected 12 hpc and fixed on glass slides using 2% paraformaldehyde. Magnification is x400. (D) *E. coli* densities (CFU/μl of hemolymph) in the hemolymph of siOBP6, siGFP and siOBP6R adults at 2 and 6 dpc (*Figure 2—source data 3*). In (B) and (D), symbols represent one hemolymph sample per group, and bars represent the median hemocyte quantity (B) or bacterial density (D) per sample. Statistical analysis = ANOVA followed by Tukey's HSD post-hoc analysis.

The following source data is available for figure 2:

**Source data 1.** Circulating hemocytes per microliter of hemolymph.
**Source data 2.** Phagocytosis by tsetse hemocytes.
**Source data 3.** Colony forming units (CFU) per microliter of hemolymph.

challenged with *E. coli*, the majority (92%) of siOBP6 adults perished after receiving a 'clean wound' to their thorax. Conversely, significantly fewer siGFP (8%) and siOBP6R (16%) controls died following this treatment (*Figure 3A*).

Adult siOBP6 tsetse perished after they were pricked with a clean needle while their siGFP and siOBP6R counterparts survived. We monitored the wound site of a select number of flies from each

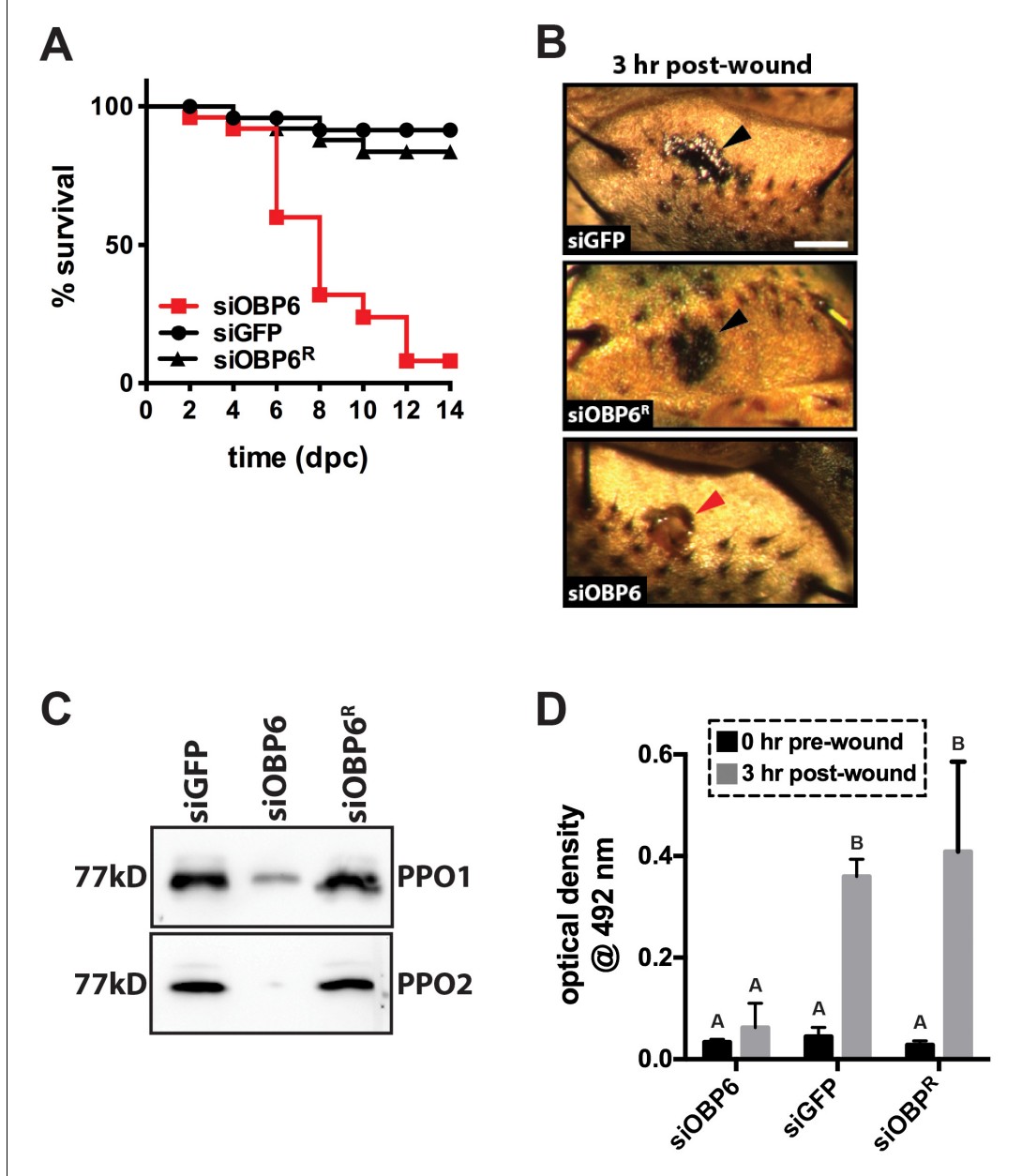

**Figure 3.** *Obp6* mediates the melanization cascade in adult tsetse. (**A**) Survival following administration of clean wounds to the thoracic cuticle of siOBP6, siGFP and siOBP6[R] adults. Survival assays were performed in triplicate, using 25 flies per replicate. Red curve depicts a statistically significant difference in infection outcome (p<0.0001, log-rank test). (**B**) A representative micrograph of the cuticle of siRNA treated adults 3 hr post-wounding (hpw) with a clean needle. Melanin deposited at the wound site of siGFP and siOBP6[R] controls, and hemolymph exudate from a siOBP6 treatment individual, are identified by black and red arrowheads, respectively. Scale bar = 500 µm. Experiment was performed using four distinct flies per group (*Figure 3—source data 1*). (**C**) Quantitation of PPO1 and PPO2 in the hemolymph of siOBP6, siGFP and siOBP6[R] adults three hpw with a clean needle. Shown is a representative Western blot analysis using *Drosophila* anti-PPO1 and anti-PPO2 antibodies. 8 µl of pooled hemolymph was run per gel lane. Hemolymph was collected and pooled from four individuals from each group. Western blots were repeated in triplicate [*Figure 3—source data 2* (for PPO1 westerns) and *Figure 3—source data 3* (for PPO2 westerns)]. (**D**) PO activity in the hemolymph of siOBP6, siGFP and siOBP6[R] adults at 0 and 3 hpw with a clean needle. *n* = 5 biological replicates per group per time point for pre-wound readings, and *n* = 8 biological replicates per group per time point for post-wound readings. Data are presented as mean ± SEM. Bars with different letters indicate a statistically significant difference between pre- and post-wound values (specific *p* values are listed in the *Figure 3—source data 4*). Statistical test = 2 way ANOVA followed by Tukey's HSD post-hoc analysis.

The following source data is available for figure 3:

*Figure 3 continued on next page*

*Figure 3 continued*

**Source data 1.** Melanin deposition at tsetse cuticular wound sites.
**Source data 2.** Tsetse prophenoloxidase 1 (PPO1) western blots.
**Source data 3.** Tsetse prophenoloxidase 2 (PPO2) western blots.
**Source data 4.** Tsetse phenoloxidase (PO) assays.

of these groups to determine if melanin was deposited at this location. Three hours post-treatment, melanin was observed at the wound site of siGFP and siOBP6[R] tsetse. Conversely, no melanin was present at the wound site of siOBP6 individuals at this time (*Figure 3B*), and, as the wound never fully healed, hemolymph continued to slowly exude from these flies for the entirety of the 2-week experimental period.

Melanin deposition represents the end product of a complex biochemical cascade, several steps of which are catalyzed by the enzyme phenol oxidase (PO; *Eleftherianos and Revenis, 2011* ). Because toxic intermediates are produced as a byproduct of melanin production, catalytic PO is usually synthesized as an inactive zymogen called prophenoloxidase (PPO; *Tang, 2009*). We quantified PPO levels in hemolymph collected from siOBP6, siGFP and siOBP6[R] flies to determine if the different wound melanization phenotypes we observed reflected different quantities of this enzyme in the hemolymph of treatment versus control individuals. Western blots using anti-PPO1 and anti-PPO2 antibodies revealed that siOBP6 adults produced significantly less of these proteins than did age-matched siGFP and siOBP6[R] individuals (*Figure 3C*). We next employed a L-DOPA assay to measure PO activity in hemolymph collected from siOBP6, siGFP and siOBP6[R] adults at 0 and 3 hr after subjection to a clean needle wound. We observed a 9.0-fold and 13.7-fold increase in PO activity in siGFP and siOBP6[R] adults, respectively, 3 hr after cuticle penetration. Conversely, PO activity only increased 2-fold in clean wounded siOBP6 adults over the same time frame (*Figure 3D*). Taken together these results indicate that when intrauterine tsetse larvae express reduced levels of *obp6* they present a dysfunctional melanization cascade during adulthood.

## *Obp6* mediates the crystal cell production pathway in larval tsetse

In *Drosophila* larvae, a specific subset of hemocytes called crystal cells produce the majority of PPO. Upon immunological stimulation, crystals cells rupture and release PPO into the hemolymph where enzymes convert it into PO that subsequently catalyzes melanin synthesis (*Honti et al., 2014*). *Drosophila* hemocytes originate in the fly's embryonic and larval lymph gland. This tissue is resorbed during metamorphosis, and evidence of prolific and prolonged de novo production of hemocytes in adult flies does not exist (*Grigorian and Hartenstein, 2013*). Assuming a similar situation occurs in tsetse, the absence of cuticular melanization in siOBP6 adults following wounding with a clean needle may reflect abnormal crystal cell development during embryogenesis and/or intrauterine larvagenesis. To investigate this hypothesis, we quantified this cell subtype in siRNA treated larvae by subjecting third instar individuals to a 65°C heat shock for 10 min. In *Drosophila* this treatment induces spontaneous activation of PPO in crystal cells, which are then visible as black melanotic spots on the larval cuticle (*Binggeli et al., 2014*). Following this treatment, we counted 7 ($\pm$2.0), 58 ($\pm$4.7) and 52 ($\pm$5.7) melanized spots on siOBP6, siGFP and siOBP6[R] larvae, respectively (*Figure 4A*). These findings suggest that siOBP6 larvae house either a reduced number of crystal cells, or that these cells exhibit a dysfunctional PPO pathway.

In *Drosophila's* lymph gland, prohemocytes express the GATA factor *serpent*, and these cells subsequently differentiate into either functionally mature plasmatocyte (phagocytic hemocytes) or crystal cell lineages via induction of the hematopoietic transcription factors *glial cells missing* or *lozenge*, respectively (*Lebestky et al., 2000*; *Ferjoux et al., 2007*). Thus, *serpent* serves as an efficacious marker for determining the presence of hemocyte progenitors, while *lozenge* expression reflects differentiation of a pool of these precursors into functional members of the crystal cell lineage (*Binggeli et al., 2014*).

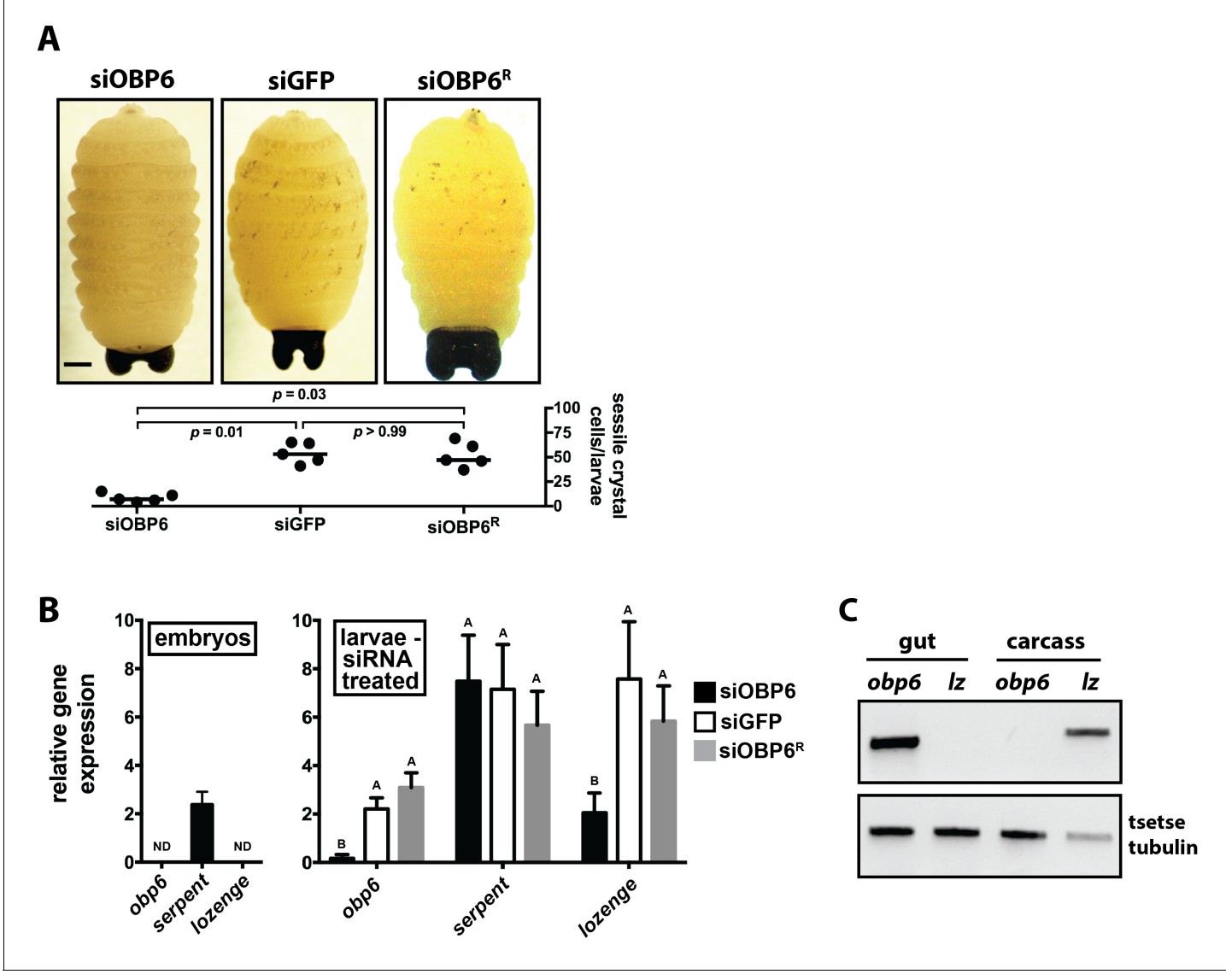

**Figure 4.** *Obp6* expression in the gut of larval tsetse is an integral component of the systemic pathway that actuates crystal cell production. (**A**) Representative micrograph depicting spontaneous PPO activation in early third instar siGFP, siOBP6 and siOBP6[R] tsetse larvae following subjection to a 10 min heat shock at 65°C. Experiment was repeated using one larvae from five distinct moms from each group. Melanotic spots were quantitated microscopically. Statistical analysis = Kruskal-Wallis test followed by Dunn's post-hoc analysis (**Figure 4—source data 1**). (**B**) RT-qPCR analysis of *obp6*, *serpent* and *lozenge* expression in embryos prior to maternal treatment with siRNA, and in siOBP6, siGFP and siOBP6[R] tsetse larvae from siRNA treated moms. Embryo replicates (*n* = 5) contain three embryos, larval replicates (*n* = 7 for siOBP6, *n* = 5 for siGFP and *n* = 6 for siOBP6[R]) contain a mixture of four first and second instar larvae. ND, not detectable. Data are presented as mean ± SEM. Bars with different letters indicate a statistically significant difference between samples (specific *p* values for larval samples are listed in the **Figure 4—source data 2**). Statistical analysis = 2 way ANOVA followed by Tukey's HSD post-hoc analysis. (**C**) Representative image of *obp6* and *lozenge* spatial expression patterns, determined using semi-quantitative RT-PCR, in the gut and carcass of second instar *Gmm*[WT] larvae. Experiment was repeated using guts and carcasses from five distinct individuals (**Figure 4—source data 3**).

The following source data is available for figure 4:

**Source data 1.** Sessile crystal abundance in larval tsetse.

**Source data 2.** Relative obp6, serpent and lozenge gene expression in tsetse embryoes and larvae.

**Source data 3.** Tissue distribution of obp6 and lozenge expression in tsetse larvae.

In an effort to determine why siOBP6 larvae present significantly fewer melanotic spots than do their siGFP and siOBP6[R] counterparts, we quantified transcript abundance of *obp6*, *serpent* and *lozenge* in tsetse embryos prior to maternal inoculation with siRNA, and then in first and second instar larvae from the three siRNA treated groups. We observed that *obp6* and *lozenge* transcripts were undetectable during embryonic development, while *serpent* expression remained unchanged across groups (*Figure 4B*). This *serpent* expression profile presented by tsetse embryos mirrors the presence of prohemocytes, while the absence of *lozenge* transcripts in individuals of this developmental stage suggests that crystal cell differentiation has yet to commence. Subsequent analysis of larval offspring from siRNA treated moms revealed that *obp6* and *lozenge* expression was significantly reduced in siOBP6 compared to siGFP and siOBP6[R] larvae, while *serpent* expression was similar in all individuals tested (*Figure 5B*). S*erpent* expression by siOBP6 larvae indicates maintenance of the hemocyte precursor pool throughout development of immature stages. However, fewer of these progenitors likely become crystal cells because siOBP6 larvae express significantly reduced levels of the *lozenge* transcription factor that actuates the differentiation process.

Intestinal microbiota can exert their physiological influence at local epithelial surfaces or peripheral tissues (*Round and Mazmanian, 2009*; *Clarke, 2014a*). To determine the spatial dynamics of *obp6* mediated induction of tsetse hematopoiesis, we analyzed the expression pattern of this gene, and *lozenge*, in gut and carcass tissues of second instar larvae. We detected *obp6* and *lozenge* transcripts only in the larval gut and carcass, respectively (*Figure 4C*). These findings indicate that *Wigglesworthia* stimulates local expression of *obp6*, and that tsetse's hematopoietic niche is likely not attached to the larval gut. Thus, symbiont-induced *obp6* regulates tsetse hematopoiesis on a systemic level.

## Obp28a, *Drosophila*'s orthologue of tsetse *obp6*, is also symbiont regulated and functionally conserved

Hematopoietic signaling pathways and their transcriptional regulators within the niche are functionally conserved across many animal taxa (*Evans et al., 2003*; *Hartenstein, 2006*; *Makhijani and Brückner, 2012*). However, little is known about the evolutionary conservation of upstream, extra-niche factors that induce blood cell lineage commitment. Thus, for comparative purposes, we investigated whether *Drosophila*'s indigenous microbiota regulate crystal cell differentiation, and thus the melanization response, within their host. To do so we measured expression levels of *obp28a* (which is orthologous to tsetse *obp6* based on sequence similarity; OrthoDB; *Kriventseva et al., 2015* ) and *lozenge* in conventionally reared and axenic (reared in the absence of their indigenous microbiota) *Drosophila* larvae. We noted that conventionally reared *w[1118]* and *Oregon-R* larvae expressed significantly more *obp28a* and *lozenge* transcripts than did their axenic counterparts (*Figure 5A*). Additionally, conventionally reared larvae and adults presented more cuticular sessile crystal cells (following heat shock) and produced more PPO, respectively, than did age-matched axenic individuals (*Figure 5B*, *Figure 5—figure supplement 1*).

We next investigated the wound healing phenotype of both a *Drosophila* RNAi line (*GAL4/UAS-obp28a RNAi*) that expresses reduced levels of *obp28a* and an *obp28a* mutant line (*Obp28a[-]*). Following thoracic injury with a clean needle, we observed that *Drosophila tub-GAL4/UAS-obp28a RNAi* and *Obp28a[-]* adults perished significantly faster than did control individuals [*tub-GAL4* driver, *UAS-obp28a RNAi* and *wCs* (mutant) progenitor lines; *Figure 5C*]. Additionally, at 6 hr post-injury, melanin had deposited at the wound site of control *Drosophila* but not their *obp28a* knockdown or knockout counterparts (*Figure 5D*). Collectively these findings suggest that OBP-mediated hematopoiesis represents an evolutionarily conserved mechanism that benefits these flies by preventing dehydration and/or exposure to opportunistic infections with environmental microbes following cuticular injury.

## Discussion

Tsetse flies must house their maternally transmitted enteric symbionts during larval development in order to present a functional cellular immune system as adults. In the absence of these microbes, tsetse larvae express reduced levels of the GATA and RUNX transcription factors *serpent* and *lozenge*. This inhibition prevents the production of blood cell progenitors and thus the differentiation of phagocyte and crystal cell lineages (*Weiss et al., 2011*, *2012*). Herein, we characterize an

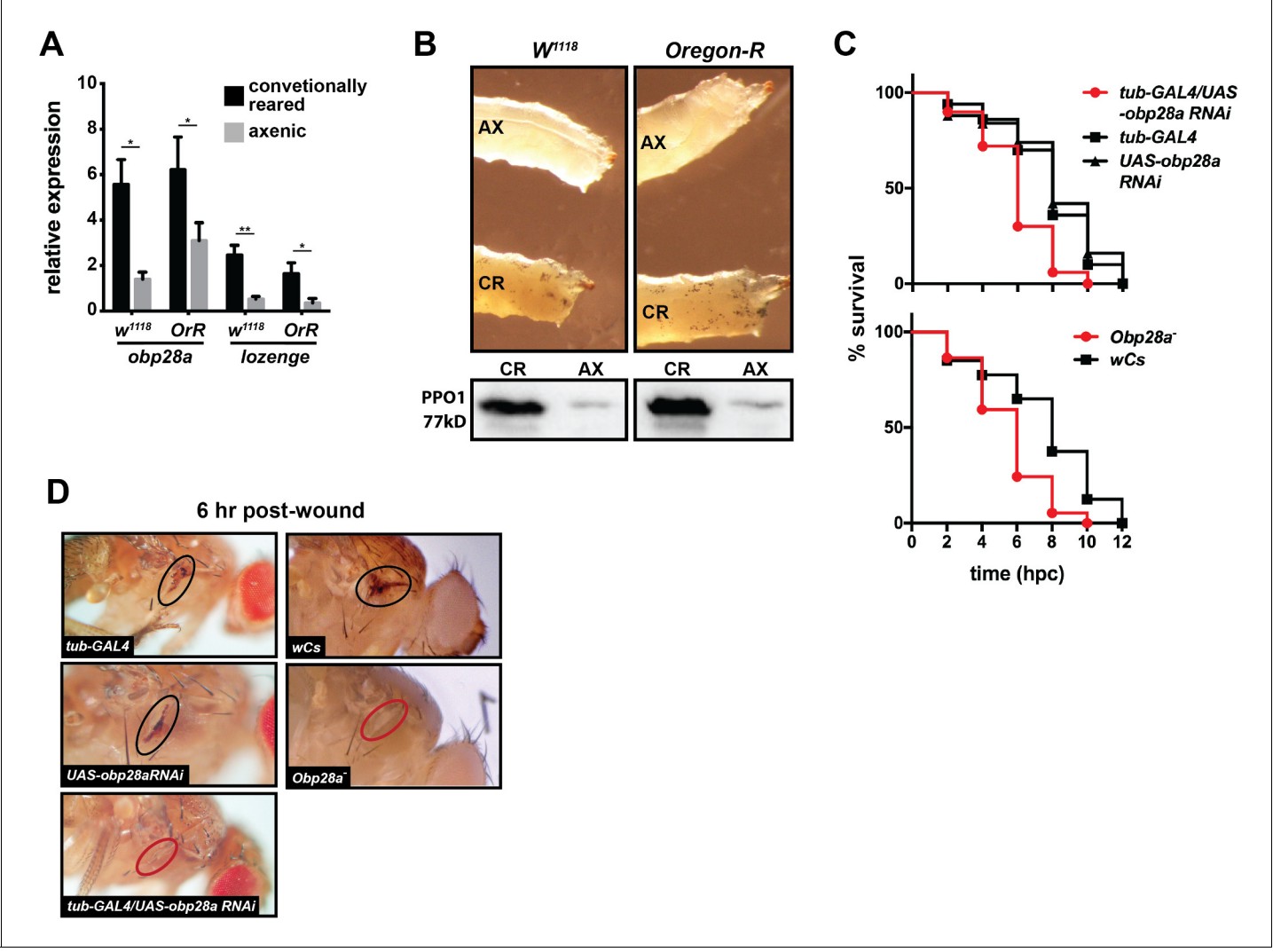

**Figure 5.** *Drosophila's* indigenous microbiota actuates larval hematopoietic pathways and thus functionality of the adult melanization response. (**A**) Relative expression of *obp28a* and *lozenge* in conventionally reared (CR) and axenic (AX) *Oregon-R* and *w1118 Drosophila* larvae. n = 9 (*Oregon-R*) and 6 (*w1118*) biological replicates per group, each containing a mixture of thirty second and early-3rd instar larvae. Data are presented as mean ± SEM. Asterisks indicate statistical significance (specific p values are listed in the ***Figure 5—source data 1***). Statistical analysis = unpaired t-tests, corrected for multiple comparisons using the Holm-Sidak method. (**B**) AX larvae house fewer sessile crystal cells, and produce less PPO, than do and CR individuals. Top panels are representative micrographs depicting spontaneous PPO activation in AX and CR *w1118* and *Oregon-R* larvae following subjection to a 10 min heat shock at 65°C. n = 5 larvae per group [***Figure 5—source data 2*** (for *Oregon-R* larvae) and ***Figure 5—source data 3*** (for *w1118* larvae)]. Bottom panels are representative Western blots using *Drosophila* anti-PPO1 antibodies. 8 μl of pooled hemolymph was run per gel lane. Hemolymph was collected and pooled from 15 individuals from each group. Western blots were repeated in triplicate (***Figure 5—source data 4***). (**C**) Survival of *obp28a* RNAi knockdown *Drosophila* (*tub-GAL4/UAS-obp28a RNAi*) and knockout (*Obp28a⁻*) adults compared to controls (*tub-GAL4, UAS-obp28a RNAi* and *wCs*) following wounding with a clean needle. Experiment was performed in triplicate, n = 50 (RNAi) and n = 45 (knockout) per group per replicate. Red curve depicts a statistically significant difference in infection outcomes [p<0.0001 (RNAi) and p=0.0002 (knockout), log-rank test]. (**D**) A representative micrograph of the cuticle of *obp28a* knockdown, knockout and control *Drosophila* adults six hpw with a clean needle. Experiment was performed using two distinct experimental fly cohorts [n = 4 flies per group per experiment; ***Figure 5—source data 5*** (for RNAi flies) and ***Figure 5—source data 6*** (for deletion mutants)]. Wounds on the cuticle of control (melanized) and *obp28a* knockdown and knockout individuals (not melanized) are identified with black and red ovals, respectively.

The following source data and figure supplement are available for figure 5:

**Source data 1.** Obp28a and lozenge expression in conventionally reared and axenic w1118 and Oregon-R Drosophila.

**Source data 2.** Sessile crystal cells in conventionally reared and axenic Oregon-R Drosophila larvae.

*Figure 5 continued on next page*

*Figure 5 continued*

**Source data 3.** Sessile crystal cells in conventionally reared and axenic w1118 Drosophila larvae.

**Source data 4.** Drosophila prophenoloxidase 1 (PPO1) western blots.

**Source data 5.** Melanin deposition at Drosophila cuticular wound sites following RNAi-mediated knockdown of obp28a.

**Source data 6.** Melanin deposition at Drosophila cuticular wound sites in obp28a deletion mutants.

**Figure supplement 1.** Axenic and conventionally reared $w^{1118}$ and *Oregon-R* larvae following subjection to a 10 min heat shock at 65°C.

obligate symbiont regulated tsetse gene that actuates a distinct component of the fly's hematopoietic program. Specifically, these microbes regulate *obp6* transcript abundance in developing intrauterine larvae. The encoded protein subsequently induces expression of *lozenge*, which drives a pool of larval hemocyte precursors to differentiate into functional crystal cells. When adult tsetse house a depleted population of this hemocyte subtype, which initiates the melanization cascade via the release of prophenoloxidase, cuticular wounds fail to clot, thus leaving the fly exposed to dehydration and/or infection with opportunistic environmental microbes. This phenotype is similar to that observed when *Wigglesworthia*-free flies are exposed to cuticular wounds (*Weiss et al., 2011*; *Figure 6*). We further discovered that *Drosophila's* indigenous microbiota regulate orthologous components of their host's hematopoietic program. Taken together, these findings accentuate the functional relevance of symbiotic bacteria as they relate to hematopoiesis, and detail an evolutionarily conserved component of the insect innate immune system that coordinates a cellular process essential for survival.

When tsetse and *Drosophila* larvae express reduced levels of orthologous *obp6* and *obp28a*, respectively, subsequent adults exhibit defective melanization cascades and thus perish unusually fast following wounding with a clean needle. These phenotypes result from reduced expression of hematopoietic *lozenge* and a consequently depleted population of PPO-producing crystal cells. In tsetse, *obp6* expression is tightly linked with intrauterine larval recognition of obligate *Wigglesworthia*. *Drosophila* larvae are free-living and thus do not obtain bacteria trans-generationally. Instead, female *Drosophila* lay their eggs in decaying organic matter, and immediately following eclosion, larvae begin feeding on this substrate (*Markow, 2015*; *Broderick and Lemaitre, 2012*). Thus, the gut of larval *Drosophila* is colonized by a relatively diverse population of environmentally acquired bacteria (*Chandler et al., 2011*; *Wong et al., 2013*). Axenic *Drosophila* larvae develop slower and weigh less than wild-type individuals (*Shin et al., 2011*; *Storelli et al., 2011*), and adults exhibit reduced stem cell activity and epithelial turnover (*Buchon et al., 2009*; *Broderick et al., 2014*) and can be more susceptible to enteric viral and bacterial pathogens (*Blum et al., 2013*; *Sansone et al., 2015*). The immune-compromised phenotypes exhibited by axenic adult *Drosophila* may reflect a developmental deficiency that, similar to the situation in tsetse, occurs when larval stages mature in the absence of symbiotic bacteria. This scenario would suggest that *Drosophila's* indigenous microbiota also influences hematopoietic processes via a mechanism homologous to *Wigglesworthia's* influence on tsetse. These convergent mechanisms, which control crucial immune phenotypes that are coordinated by evolutionarily conserved genes and regulatory pathways, exist despite the divergent nature of these host-symbiont model systems. Specifically, the tsetse-*Wigglesworthia* symbiosis originated 50–80 million years ago (*Chen et al., 1999*), and due to the bacterium's strict vertical route of transmission, is highly steadfast in nature. Conversely, *Drosophila's* relationship with its microbiota is relatively transient and highly dependent on diet and local environmental factors (*Blum et al., 2013*; *Broderick et al., 2014*). The extent to which o*bp6* and *obp28a* orthologues are functionally conserved in other arthropods remains to be determined. Furthermore, although vertebrates lack direct *obp6 and obp28a* orthologues (*Kriventseva et al., 2015*), unrelated odorant binding proteins may present similar roles in these systems.

Innate immunity is germ-line encoded and regulated by highly conserved pathways (*Rubin et al., 2000*; *Lemaitre and Hoffmann, 2007*), some of which are actuated by systemically derived signals. This type of molecular coordination is well characterized in *Drosophila*, where insulin secreted by

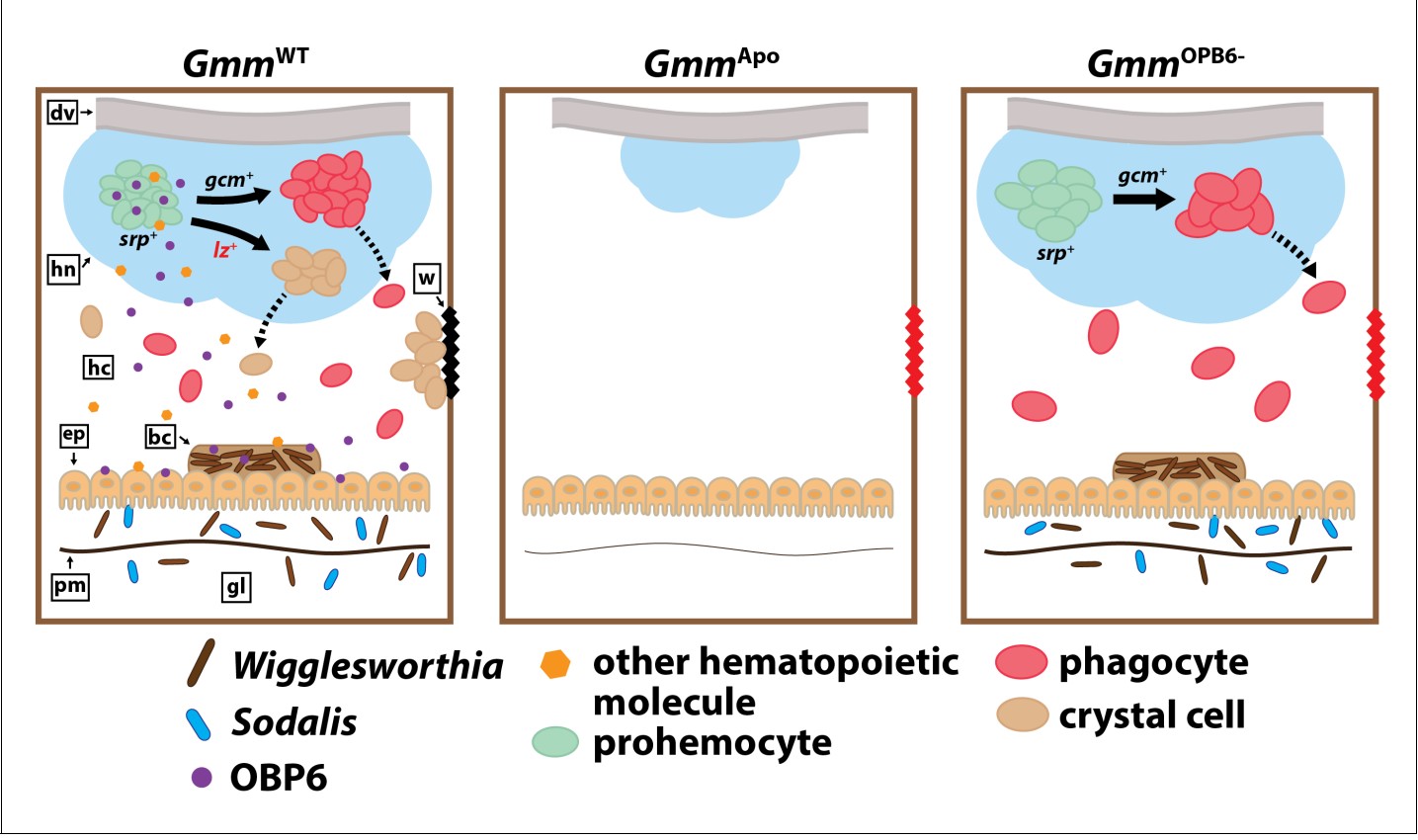

**Figure 6.** Model illustrating the functional relationship between maternally-transmitted enteric symbionts and melanization in tsetse. $Gmm^{WT}$ larvae imbibe enteric symbiotic-containing milk gland secretions throughout their intrauterine developmental program. These bacteria colonize larval gut-associated tissues, including the bacteriome, and in doing so, induce the expression of *obp6*. OBP6 is either secreted directly into the hemolymph, or acts locally to induce expression of another unknown, (also secreted) protein. One of these molecules then acts systemically in the larval hematopoietic niche (hn) to stimulate *lozenge* (*lz*) expression in a small proportion of *serpent* (*srp*) expressing prohemocytes. These cells then become PPO-producing crystal cells [remaining prohemocytes become phagocytes after expressing *glial cells missing* (*gcm*)]. Finally, crystal cells are expelled from the hn, where they circulate in the hemolymph and are available to produce wound-healing melanin. Larvae that develop in the absence of symbiotic bacteria ($Gmm^{Apo}$) fail to produce any hemocytes, while those that develop in the presence of reduced *obp6* transcript abundance ($Gmm^{OPB6-}$) fail to express *lozenge* and thus likely fail to generate crystal cells. dv, dorsal vessel; hc, hemocoel; w, wound; ep, epithelial cells of midgut; bc, bacteriome; pm, peritrophic matrix; gl, gut lumen.

insulin-producing cells in the brain, and essential amino acids emanating from the gut, promote Wingless signaling that maintains blood cell progenitors in the hematopoietic niche (*Shim et al., 2012*). Smell also contributes to progenitor maintenance in larval *Drosophila*. Specifically, olfactory receptor neurons, stimulated by the detection of small food-derived volatile molecules, secrete GABA into the larval hemocoel where it subsequently binds to blood cell progenitors in the lymph gland. This interaction increases cytosolic calcium concentrations that are necessary and sufficient to maintain the progenitor population. When larval *Drosophila* are reared on an odor-free diet, they fail to retain a pool of these cells (*Shim et al., 2013a*). These findings highlight the intriguing association between smell and homeostasis of innate immune-related activities. For the experiments performed herein, *Drosophila* and tsetse were reared in the absence of indigenous microbes, but on diets (sterilized) and in environments that emitted normal food odors. Under these conditions both flies presented dysfunctional hematopoietic programs, indicating that microbe-derived factors may also influence hematopoiesis. While the exact chemical structure of these molecules is currently unknown, they could take the form of microbe-associated molecular patterns (MAMPs), including bacterial cell wall components such as polysaccharide A, peptidoglycan and Nod-like receptor ligand, which in mammals actuate differentiation of T cell lineages (*Mazmanian et al., 2005*), enhance bone marrow-

derived neutrophil killing (*Clarke et al., 2010*) and stimulate macrophage activity in lung tissues (*Clarke, 2014b*; *Gauguet et al., 2015*), respectively. Additionally, microbial metabolites mediate regulatory T cell abundance in colonic tissues, as germ-free mice that lack gut microbiota-derived fatty acids present fewer of this immune cell type (*Smith et al., 2013*; *Furusawa et al., 2013*). These findings accentuate the concept that general metabolic defects associated with dysbiosis can give rise to cellular immunity-related pathologies (*Norata et al., 2015*), including impairment of hematopoietic programs (*Shim et al., 2012*, *2013b*).

Bacteria also release a wide variety of volatile compounds (*Audrain et al., 2015*), some of which, following chemosensory detection, influence animal immune phenotypes. A situation of this nature occurs when adult *Drosophila* detect geosmin, which is a microbial odorant that signals the presence of harmful microbes. This stimulus induces expression of olfactory receptor *Or56a* in olfactory sensory neurons, which leads to an aversion behavior, reduction in the activity of other olfactory pathways, and inhibition of positive chemotaxis, oviposition and feeding behavior (*Stensmyr et al., 2012*). A similar phenomenon occurs in the nematode *Caenorhabditis elegans*, which executes a protective avoidance behavior following chemosensory detection of the aromatic compounds phenazine-1-carboxamide and pyochelin that are produced by pathogenic *Pseudomonas aeruginosa* (*Meisel et al., 2014*). *Drosophila* detects and prefers specific odors produced by members of its microbiome (*Fischer et al., 2016*). As such, odorant molecules derived from tsetse and *Drosophila* enteric microbes could modulate the expression of *obp6* and *obp28a* in the gut of larval flies, thus serving as the signal that activates hematopoiesis.

In larval tsetse *Wigglesworthia* is found exclusively within the gut-bacteriome axis (*Balmand et al., 2013*), and o*bp6* expression is restricted to this same environment. Conversely, *lozenge* transcripts are found solely in the carcass, indicating that the fly's hematopoietic niche is likely not attached to the gut. The spatial expression pattern of these genes indicates that OBP6 exerts a systemic effect on tsetse hematopoiesis, and this process could occur in one of two ways. First, this protein could act locally at the mucosal interface to induce the production and secretion of a distinct immuno-stimulatory molecule that subsequently acts in tsetse's hematopoietic niche. A mechanism of this nature occurs in mice, where enteric symbiont-derived molecules induce group three innate lymphoid cells in the intestine to increase production of the cytokine interleukin-17 (IL-17). Circulating IL-17 subsequently induces production of peripheral granulocyte colony-stimulating factor, which actuates neutrophil differentiation in bone marrow tissues (*Deshmukh et al., 2014*). Insect OBPs are widely believed to serve as vehicles that carry odorant molecules from sensory sensillum pores to corresponding dendritic odorant receptors (*Leal, 2013*). Based on this definition, tsetse OBP6 could also serve a carrier-like function following translocation from the fly's gut into its circulatory system. More specifically, this protein could complex with another hematopoietic molecule produced either in tsetse's gut or extra-intestinally, and then both together, or the latter molecule alone, stimulate hemocyte development in the hematopoietic niche. Notably, the function of insect OBPs was defined based on experiments performed in the chemosensory apparatus, and the role of these proteins in other insect tissue types is largely unknown. Interestingly, a recent study exploring the roles of OBPs in *Drosophila* sensilla found that deletion of *obp28a* did not reduce the magnitude of fly olfactory responses, suggesting a novel role for the encoded protein (*Larter et al., 2016*). As such, tsetse OBP6 could also function unconventionally by directly stimulating hematopoiesis following secretion into the larval hemocoel. Further studies are required to determine the specific cell type within larval tsetse's gut and/or bacteriome that express *obp6*. Additionally, identification of OBP6 target tissues will provide valuable insight into the immuno-stimulatory function of this protein.

Finally, we speculate that our tsetse fly related findings reported herein are directly relevant to the transmission of vector-borne pathogens. Melanization is a cellular immune response that occurs across invertebrate taxa (*Cerenius and Söderhäll, 2011*), and this mechanism influences the transmission of several pathogens through their respective insect vectors (*Collins et al., 1986*; *Zou et al., 2008*; *Bartholomay, 2014*). Thus, an increased understanding of the physiological processes that regulate melanization may have translational implications pertinent to the development of pathogen-refractory insects.

## Materials and methods

### Fly lines and bacteria

*G. morsitans morsitans* were maintained in Yale's insectary at 24°C with 50–55% relative humidity. All flies received defibrinated bovine blood (Hemostat Laboratories) every 48 hr through an artificial membrane feeding system. Aposymbiotic tsetse larvae (*Gmm*^Apo^) were derived from females fed a diet supplemented with tetracycline (20 μg per ml of blood) to clear their indigenous microbiota, and yeast extract (1% w/v) to rescue the sterile phenotype associated with the absence of *Wigglesworthia* (*Alam et al., 2011*). Thus, *Gmm*^Apo^ offspring developed in the absence of all symbiotic bacteria. Embryos and 1^st^ and second instar larvae for all experimental groups were age-matched by taking individual samples from pregnant females undergoing their second gonotrophic cycle (*Attardo et al., 2014*).

Axenic *Drosophila* larvae (*Oregon-R* and *w^1118^* strains) were generated as described previously (*Broderick et al., 2014*). *Drosophila tub-GAL4* driver and *UAS-obp28a RNAi* progenitor lines were crossed, and resulting *tub-GAL4/UAS-obp28a RNAi* F1 offspring were used for experiments (*Dietzl et al., 2007*). Anti-*obp28a* RNAi target gene specificity, knockdown efficacy and accompanying phenotypic characterization, were determined previously (*Swarup et al., 2011*). Obp28a deletion mutants (*Obp28a^-^*) were generated from progenitor flies (*CAS-0003*; *Kondo and Ueda, 2013*) using the CRISPR-Cas9 system, and backcrossed to *w Canton-S* (*wCs*) for five generations (*Larter et al., 2016*). Flies were maintained on a cornmeal-yeast-agar medium (per liter of water: 50 g inactivated yeast, 70 g maize flour, 6 g agar, and 40 g of dextrose) at 25°C in ambient humidity.

GFP-expressing *E. coli* K12 (*recE. coli*~GFP~) were produced via electroporation with pGFP-UV plasmid DNA (Clontech).

### Transcriptomics

First and second instar tsetse larvae (*n* = 5 of each) were collected from two distinct cohorts of pregnant *Gmm*^WT^ and *Gmm*^Apo^ females 48 hr post-feeding. Total RNA from the above-mentioned tsetse larvae was extracted, DNAse treated and purified as described previously (*Benoit et al., 2014*). RNA-seq libraries were constructed using polyadenylated RNA and standard Illumina RNA-seq protocols. Libraries were sequenced at the McDonnel Genome Institute (Washington University). Read files have been deposited in the NCBI BioProject database (ID# PRJNA309164).

FastQC analyses were performed on the RNA-seq datasets to assess read quality. Low quality reads and sequencing adaptors were removed with Trimmomatic (*Bolger et al., 2014*). Transcript expression levels were determined using CLC Genomics Workbench (CLC Bio, Cambridge, MA). Briefly, RNA-seq datasets were mapped directly to the tsetse fly genome (*International Glossina Genome Initiative, 2014*) with an algorithm that allowed only two mismatches and a maximum of 10 hits per read. Transcripts per million (TPM) was used as a proxy for gene expression. The predicted gene set associated with the genome was version 1.1 obtained from Vectorbase (*Giraldo-Calderón et al., 2015*). The following samples were compared: *Gmm*^WT^ larvae against *Gmm*^Apo^ larvae [Sequence Read Archive (SRA) IDs SRR3107831-SRR3107834 (BioProject ID PRJNA309164)], and *Gmm*^WT^ larvae against *Gmm*^WT^ male and female adults (BioProject IDs PRJNA295435 and PRJNA205861, respectively; *Scolari et al., 2016*; *Benoit et al., 2014*). SRA and BioProject sequence data is available at the NCBI website (http://www.ncbi.nlm.nih.gov/sra and http://www.ncbi.nlm.nih.gov/bioproject, respectively). Relative fold differences in gene expression between samples were determined as a ratio of each TPM. Significance was determined via Baggerly's test followed by a false detection rate at $p < 0.01$ (*Baggerly et al., 2003*). In conjunction with the above-described analyses, we conducted a de novo assembly of the larval transcriptomes using Trinity (*Grabherr et al., 2011*), followed by RNA-seq analyses to identify additional differentially expressed genes of interest between *Gmm*^WT^ and *Gmm*^Apo^ individuals. No additional targets were identified by this secondary analysis. Thus, subsequent functional studies focused on results obtained from the predicted genomic gene set.

Predicted genes were annotated using tblastx, with an E-value cut-off of $1e^{-10}$ and bit score of 200, to a previously annotated *Glossina* transcriptome (*Benoit et al., 2014*). Another comparative analysis, using the same parameters, was performed with annotated protein sequences from *D. melanogaster* and *Pediculus humanus* from FlyBase and Vectorbase, respectively. Blast2GO was utilized

to identify specific gene ontology (GO) terms that were enriched between treatments based on a Fisher's Exact Test (*Conesa et al., 2005*). Specific GO-based functional categories were developed based on comparison with associated *D. melanogaster* genes acquired from Flybase (*Marygold et al., 2013*; *Attrill et al., 2016*). These categories included those involved in B vitamin metabolism, hematopoiesis, midgut development, larval development, immunity, organismal growth and chitin associated. For these category assignments *Drosophila* and *G. morsitans* gene sets were compared, and a functional match was considered valid if the E-value was below $10^{-40}$ and the bit score was over 200. Enrichment for a specific GOC associated with each sample was determined with a Fisher's Exact Test. Specific analyses of the *odorant binding protein* genes were accomplished by obtaining predicted models for these genes from the *G. morsitans* genome.

## RNA interference

A cartoon summarizing temporal aspects of RNAi and subsequent functional experiments is shown in *Supplementary file 3*. Two cohorts (*n* = 150 individuals per group) of virgin female tsetse were mated three days post-eclosion (dpe), and embryos (*n* = 3 in each of seven biological replicates) were collected five days later from a subset of individuals to obtain baseline *serpent* (*srp*) and *lozenge* (*lz*) expression values. Subsequently, pregnant female flies were subjected to thoracic microinjection (using glass needles and a Narashige IM300 micro-injector) with either anti-*obp6* (treatment) or anti-*gfp* (control) siRNAs (siRNA sequences listed in *Supplementary file 5*) on days 8 and 11 post-mating. This window of time post-mating was chosen in an effort to expose feeding first and/or second instar larval tsetse to siRNAs (generated by Integrated DNA Technologies, Coralville, IA) taken up by the milk gland following inoculation into the maternal hemocoel. Anti-*obp6* siRNA was coupled to a Cy3 dye (siOBP6$_{Cy3}$) to track nucleic acid dissemination through the maternal hemocoel and into the developing intrauterine larvae. siOBP6$_{Cy3}$ larvae were rigorously washed in PBS prior to stimulation with UV light to ensure Cy3 labeled siRNA was removed from the cuticular surface. All anti-*obp6* and anti-*gfp* siRNA treated moms and their larval and adult offspring are designated 'siOBP6' and 'siGFP', respectively, throughout this study.

siRNA target specificity was confirmed in silico at VectorBase via BLAST analysis against a *G. morsitans* RNA-seq library, and a complete set of tsetse genomic scaffolds (both available on the Vector-Base website; www.vectorbase.org).

## PCR assays

Real time quantitative real-time PCR (RT-qPCR) was performed as described previously (*Weiss et al., 2012*). All RT-qPCR results were normalized relative to tsetse's constitutively expressed *β-tubulin* or *Drosophila's Rpl32* gene (determined from each corresponding sample). Replicate numbers and sample sizes are presented on figures or in their corresponding legends.

For semi-quantitative reverse transcription PCR (RT-PCR) analysis, second instar individuals were removed from pregnant females and an incision was made the length of the larval cuticle. The larval gut, which rapidly exudes following cuticular incision, and the corresponding carcass, were collected separately in PBS. RNA and cDNA were made from larval gut and carcass tissues as described previously (*Weiss et al., 2012*). Tsetse *β-tubulin* was used as a loading control.

RT-qPCR and RT-PCR primers are listed in *Supplementary file 6*. Primer target specificity was confirmed in silico at VectorBase and FlyBase via BLAST analysis against *G. morsitans* and *D. melanogaster* RNA-seq libraries.

## Infection and wounding experiments

siGFP and siOBP6 adult tsetse were subjected to systemic challenge with *recE. coli*$_{GFP}$ during adulthood. Percent survival was subsequently monitored over a two week period. For *E. coli* infections, tsetse were anesthetized on ice and subsequently injected with $5 \times 10^2$ colony forming units (CFU) of live bacterial cells using glass needles and a Narashige IM300 micro-injector.

'Clean' wounds were administered to siGFP, siOBP6 and siOBP6$^R$ adult tsetse, and *tub-GAL4*, *UAS-obp28a RNAi*, *tub-GAL4/UAS-obp28a RNAi*, *Obp28*$^-$ and *wCs* adult *Drosophila,* by pricking individual flies in the thorax with a heat sterilized glass needle. Injured tsetse were housed under normal insectary conditions, while injured *Drosophila* were maintained in a desiccated environment with no access to food or water. All tsetse survival experiments were performed in triplicate, using

25 flies per replicate. All *Drosophila* survival experiments were performed in triplicate using 50 flies per replicate.

## Hemolymph extraction and hemocyte quantification and visualization

Hemolymph was collected by removing a fly leg with forceps and exerting gentle pressure on the abdomen, thus causing a hemolymph droplet to exude from the neck. Determination of circulating hemocyte abundance was performed using a Bright-Line hemocytometer.

Hemocyte phagocytic capacity of siOBP6 and siGFP adults was determined by injecting individuals with $5 \times 10^2$ CFU of live rec$E. coli_{GFP}$. Six hours post-inoculation, hemolymph was collected from three individuals and hemocytes monitored for the presence of engulfed GFP-expressing bacterial cells. Hemolymph samples were fixed on glass microscope slides via a 2 min incubation in 2% paraformaldehyde and then overlaid with VectaShield hard set mounting medium containing DAPI (Vector Laboratories, Burlingame, CA). Cells were visualized using a Zeiss Axioscope microscope.

## Bacterial quantification

To quantify rec $E. coli_{GFP}$ in siOBP6 and siGFP individuals at 2 and 6 days post-challenge ($n = 5$ per siRNA treatment per time point), 3 µl of hemolymph was serially diluted in 0.85% NaCl and plated on LB/agar supplemented with ampicillin (50 µg/ml). CFU per plate were counted manually.

## Prophenoloxidase and phenoloxidase activity assays

Western blots were performed in triplicate, with each replicate containing 8 µl of hemolymph. For tsetse, 2 µl of hemolymph (collected by removing a leg at the proximal joint nearest the thorax) was pooled (and then immediately frozen) from four flies per group 3 hr post-wounding (hpw) with a clean glass needle. Adult *Drosophila* ($n = 75$ per group) were thoracically wounded with a sterilized tungsten needle. Three hpw, flies were chilled on ice and placed into Zymo-Spin IV columns ($n = 15$ flies per column; Zymo Research, Irvine, CA) preloaded with 0.5 mm glass beads (Scientific Industries, Bohemia, NY). Columns were centrifuged at 4°C for 15 min and hemolymph pooled in the column collection tube was frozen. Denatured (100°C for 5 min in protein loading buffer) hemolymph extracts were separated on a 10% polyacrylamide gel, transferred to nitrocellulose membranes, blocked with 3% bovine serum albumin (prepared in PBST buffer) for 1 hr at room temperature and incubated overnight at 4°C with rabbit anti-PPO1 or anti-PPO2 (generated against recombinant *Drosophila* PPO1 and PPO2, respectively; *Nam et al., 2012*) antibodies at a 1:1500 dilution. Blots were subsequently probed with an HRP conjugated goat anti-rabbit 2° antibody (BioRad, Hercules, CA), and PPO protein bands were visualized using a SuperSignal West Pico Chemiluminescent Substrate kit according to the manufacturer's (Thermo Scientific, Waltham, MA) protocol.

L-3,4-dihydroxyphenylalanine (L-DOPA; Sigma-Aldrich, St. Louis, MO) assays (performed as described in *Perdomo-Morales et al., 2007*; *Binggeli et al., 2014*) were used to quantify enzymatic phenoloxidase (PO) activity in 3 µl of hemolymph collected from siOBP6, siGFP and siOBP6$^R$ individuals immediately (0 hr; $n = 5$ individuals per group at this time point) and 3 hr post-wounding ($n = 8$ individuals per group at this time point) with a clean needle. Enzymatic activity of tsetse's melanization cascade was arrested by adding protease inhibitor at the time of hemolymph collection. Thus, values reflect in vivo PO at this time point. Values are represented as the mean (±SEM).

## Larval crystal cell analysis

In *Drosophila*, sessile crystal cells attached to the hemocoelic side of the larval cuticle can be visualized as dark spots following spontaneous activation of PPO (*Binggeli et al., 2014*). This phenotype was induced in tsetse and *Drosophila* larvae by heating individuals to 65°C for 10 min. PPO spots were quantified visually using a dissecting microscope (Zeiss Discovery) equipped with a digital camera (Zeiss AxioCam MRc 5).

## Bacterial complementation

Three cohorts ($n = 25$ individuals/group) of pregnant female tsetse were fed a diet containing tetracycline (100 µg/ml of blood) every other day for 10 days to eliminate all indigenous bacteria. All blood meals (three per week), throughout the course of the entire experiment, also contained vitamin-rich yeast extract (1% w/v) to restore fertility associated with the absence of *Wigglesworthia*

(*Weiss et al., 2012*). Ten days post-copulation, 4 cohorts of symbiont-cured females were regularly fed a diet supplemented with either 1) *Wigglesworthia*-containing bacteriome extracts (obtained by dissecting bacteriomes from $Gmm^{WT}$ females), 2) *Wigglesworthia*-free bacteriome extracts (derived from the offspring of females fed a diet supplemented with ampicillin, which results in the production of progeny that lack *Wigglesworthia* but still harbor Sodalis; *Pais et al., 2008*; *Weiss et al., 2011*), 3) *Sodalis* cell extracts (derived from *Sodalis* maintained in culture; *Hrusa et al., 2015*), and 4) bacteriome extracts harvested from aposymbiotic females. Offspring (symbiont-free) of these extract supplemented females were designated $Gmm^{bact/Wgm+}$, $Gmm^{bact/Wgm-}$, $Gmm^{Sgm+}$ and $Gmm^{bact/Apo}$, respectively. Bacteriome supplemented females were fed one tissue equivalent per four females, and $Gmm^{Sgm+}$ females were fed $4 \times 10^7$ Sodalis per ml of blood (these flies thus ingested ~$1 \times 10^6$ Sodalis each time they fed). Control cohorts consisted of wild-type ($Gmm^{WT}$) and aposymbiotic ($Gmm^{Apo}$) offspring.

RT-qPCR was used to determine if complementation with bacterial extracts plus yeast altered the expression pattern of *obp6* in aposymbiotic larvae (First and second instar) from the second gonotrophic cycle (these offspring were used to ensure that antibiotic treatment had cleared all maternal symbionts such that none were present for transmission to larvae) of symbiont-cured moms.

## Replicates and statistics

Throughout the manuscript, all replicates are 'biological', implying that data were obtained by repeating experiments using the indicated number of distinct samples. Replicates and sample sizes for all experiments are provided in the legend that corresponds to each representative figure (except for *Figure 1*, for which sample size is indicated in the 'Materials and methods section, subheading 'Transcriptomics'). Statistical significance between treatments and controls is indicated on figures or in the corresponding figure legends. Tests used to determine statistical significance are indicated in figure legends. All statistical analyses were performed using GraphPad Prism software (v.6).

## Acknowledgements

The authors thank Dr. Wesley Warren (Washington University) for sequencing of RNA-seq libraries and Dr. Won-Jae Lee (Seoul National University) for providing *Drosophila* anti-PPO1 and anti-PPO2 antibodies. We thank members of the Aksoy lab for their critical reviews of the manuscript. These studies were supported by the NIH/NIAID [research grants to Brian L Weiss (R21AI1011456), Serap Aksoy (R01AI051584) and John Carlson (R01DC02174)] and the Ambrose Monell Foundation. Jennifer S Sun acknowledges financial support from the NSF Graduate Research Fellowship Program, National Institutes of Health (NIH T32 GM007499), and the Dwight N and Noyes D Clark Scholarship Fund.

## Additional information

### Funding

| Funder | Grant reference number | Author |
| --- | --- | --- |
| National Institutes of Health | R21AI1011456 | Brian L Weiss |
| National Institutes of Health | R01AI051584 | Serap Aksoy |
| National Institutes of Health | R01DC02174 | John R Carlson |
| Ambrose Monell Foundation | | Serap Aksoy |
| National Science Foundation | Graduate Research Fellowship Program | Jennifer S Sun |
| National Institutes of Health | T32GM007499 | Jennifer S Sun |
| Dwight N and Noyes D Clark Scholarship Fund | | Jennifer S Sun |

The funders had no role in study design, data collection and interpretation, or the decision to submit the work for publication.

## Author contributions

JBB, AV, NAB, Conception and design, Acquisition of data, Analysis and interpretation of data, Drafting or revising the article; YW, performed experiments, Acquisition of data; JSS, JRC, Drafting or revising the article, Contributed unpublished essential data or reagents; SA, Conception and design, Analysis and interpretation of data, Drafting or revising the article, Contributed unpublished essential data or reagents; BLW, Conception and design, Acquisition of data, Analysis and interpretation of data, Drafting or revising the article, Contributed unpublished essential data or reagents

## Author ORCIDs

Nichole A Broderick, http://orcid.org/0000-0002-6830-9456
Jennifer S Sun, http://orcid.org/0000-0002-4274-0504
John R Carlson, http://orcid.org/0000-0002-0244-5180
Serap Aksoy, http://orcid.org/0000-0001-9941-143X
Brian L Weiss, http://orcid.org/0000-0003-1576-3556

## Additional files

### Supplementary files

• Supplementary file 1. Results of RNA-seq analysis, indicating genes that are expressed at significantly different levels (Baggerly's test followed by a false detection rate at $p < 0.01$), in $Gmm^{WT}$ and $Gmm^{Apo}$ larvae.

• Supplementary file 2. Ontogeny analysis of hematopoiesis associated genes in $Gmm^{WT}$ compared to $Gmm^{Apo}$ larvae.

• Supplementary file 3. Cartoon summarizing RNAi and subsequent functional experiments. All relevant experimental details are described in the Materials and methods, under the 'RNA interference' sub-heading.

• Supplementary file 4. Trans-generational RNAi-based depletion of $obp6$ expression in intrauterine $Gmm^{WT}$ larvae. (A) Pregnant females were intra-thoracically injected with Cy3 tagged anti-$obp6$ short interfering (si) RNAs (siOBP6$_{Cy3}$). A representative micrograph showing that siOBP6$_{Cy3}$ had disseminated throughout the maternal hemocoel by three days post-treatment (dpt; top left panel), and was present in second instar larvae from the first gonotrophic cycle (GC1; middle left panel) of these females as visualized using fluorescent illumination (bottom left panel). By 26 dpt, siOBP6$_{Cy3}$ was absent from treated moms (top right panel) and their third gonotrophic cycle (GC3) larvae (middle and bottom right panels). Five pregnant treatment and recovered females were visualized to observe siOBP6$_{Cy3}$ dissemination and transfer to larvae. (B) Effectiveness of siRNA-based $obp6$ knockdown in intrauterine second instar tsetse larvae. Relative expression of $obp6$ in second instar siGFP, siOBP6 and siOBP6$^R$ intrauterine larvae. RT-qPCR analysis was performed using larvae from two distinct experiments, each of which included 4 (siGFP and siOBP6) or 3 (siOBP6) biological replicates (each consisting of a mixture of four first and second instar larvae). All RT-qPCR results were normalized relative to tsetse's constitutively expressed $\beta$-tubulin gene (determined from each corresponding sample). Data are presented as mean of all replicates from both experiments, ± SEM. Bars with different letters indicate a statistically significant difference ($p < 0.05$) between treatments. Statistical analysis = 2 way ANOVA.

• Supplementary file 5. DsiRNAs used in this study.

• Supplementary file 6. PCR primers used in this study.

## Major datasets

The following dataset was generated:

| Author(s) | Year | Dataset title | Dataset URL | Database, license, and accessibility information |
|---|---|---|---|---|
| Joshua B Benoit, Aurélien Vigneron, Nichole A Broder-ick, Yineng Wu, Jennifer S Sun, John R Carlson, Serap Aksoy, Brian L Weiss | 2016 | Glossina morsitans strain: Yale Transcriptome or Gene expression: 1st instar tsetse fly larvae (whole organism) | https://www.ncbi.nlm.nih.gov/bioproject/?term=PRJNA309164 | Publicly available at NCBI BioProject (accession no. PRJNA309164) |

The following previously published datasets were used:

| Author(s) | Year | Dataset title | Dataset URL | Database, license, and accessibility information |
|---|---|---|---|---|
| Scolari F, Benoit JB, Michalkova V, Aksoy E, Takac P, Abd-Alla AM, Malacrida AR, Aksoy S, Attadro GM | 2016 | Glossina morsitans morsitans Male accessory gland and testes raw illumina reads | http://www.ncbi.nlm.nih.gov/bioproject/?term=PRJNA295435 | Publicly available at NCBI BioProject (accession no. PRJNA295435) |
| Ichalkova V, Krause TB, Bohova J, Zhang Q, Baumann AA, Mireji PO, Ta-kac P, Denlinger DL, Ribeiro JM, Aksoy S | 2014 | Glossina morsitans morsitans transcriptome during milk production | http://www.ncbi.nlm.nih.gov/bioproject/?term=PRJNA205861 | Publicly available at NCBI BioProject (accession no. PRJNA205861) |

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
