## [Decision Letter]

Thank you for submitting your article "Symbiont-induced odorant binding proteins mediate insect host hematopoiesis" for consideration by *eLife*. Your article has been favorably evaluated by K VijayRaghavan (Senior Editor) and three reviewers, one of whom, Bruno Lemaitre, is a member of our Board of Reviewing Editors.

Summary:

We know that resident microorganisms play a very important role in the development of the immune system of mammals, but much of the detail remains to be resolved and the significance of microorganisms for the immune system development in other animals is largely unknown. Indeed, previous research of these authors showing that the bacterial symbiont *Wigglesworthia* or a cell-free extract of this bacterium is required for normal development of hemocytes in Tsetse flies (Weiss et al. 2012) provides one of the very few studies on the role of bacteria in maturation of the immune system of any invertebrate animal. This study is important because it demonstrates key elements of the molecular mechanism of the *Wigglesworthia* effect. It is genuinely intriguing that a single insect protein, OBP6 is necessary and sufficient for the effect, and that the effect is restricted to the differentiation of one type of hemocyte, the crystal cell.

Essential revisions:

Although the experimental set-up and work is highly appreciated, the presented data are in the view of the three reviewers not waterproof to fully support the claim that 1) the intrauterine larval recognition of the obligate *Wigglesworthia* symbiont regulates the expression of the larval OBP6 and 2) this is an evolutionary conserved component of the insect innate system. The three reviewers raised three important issues, each of them should be addressed before acceptance.

1) Specificity

The authors claim that the regulation of the larval OBP6 expression and the subsequent production of the crystal cells are regulated by the obligate symbiont *Wigglesworthia*. This is based upon the sole observation that the deprived *obp6* expression in 1st / 2nd instar larvae of aposymbiotic flies (= offspring of tetracycline-treated females that lack all microbiota i.e. *Wigglesworthia, Sodalis* and *Wolbachia* present in tsetse lab colonies) is restored by the supplementation of the blood meal given to the mother flies with a *Wigglesworthia* extract. This 'extract' is actually a homogenate of the bacteriome of WT flies that is a cocktail of the *Wigglesworthia* bacterium with cellular/molecular components of the tsetse anterior midgut. This means that the presented experimental work is only suggestive for a possible involvement of *Wigglesworthia* in the larval *obp6* expression but is not conclusive. Moreover, it is not clear how *Wigglesworthia* or its components that are administered through a blood meal could finally reach the larva that is developing in the fly uterus. In previous work, the authors showed that *Wigglesworthia* cannot colonize aposymbiotic flies when administered this way (Weiss et al.2012). In previous work the authors used an ampicilline-treatment approach to have tsetse flies that produce offspring that is specifically cleared for *Wigglesworthia* but not for the other microbiota. The use of use this model for the experimental work could make the link between *Wigglesworthia*/OBP6/crystal cells more conclusive. Also, an additional experimental group where the blood meal is supplemented with the bacteriome of aposymbiotic tsetse (contain tsetse midgut components but devoid of *Wigglesworthia*) would be at least a more appropriate control group in experiment 1F.

2) A better characterization of Tsetse crystal cell

The identification of crystal cells in the tsetse fly appears to be based entirely on the number of melanized spots in the cuticle after heat shock. Previous work by different authors used microscopical analysis of crystal cells to develop and validate the reliability of this assay for *Drosophila*. Microscopical demonstration of crystal cells in untreated tsetse and their depletion in aposymbiotic or RNAi-treated tsetse would give confidence in the reliability of the assay in tsetse.

3) The part on *Drosophila* should be better documented

The part on *Drosophila* is really exciting, suggesting a conserved mechanism. Unfortunately, it is too short in the present version, and straightforward experiments should be added to reinforce the conclusion. The impact of microbiota on *obp28a* and lozenge expression should be repeated in another genetic background. The study will be strongly reinforced by the use of *obp28a* mutation (a temporally feasible with the advent of CRISPR), or at least the use of another independent RNAi to confirm the first one. In which tissues is *obp28a* expressed in larvae? The status of crystal cells could be tested by cooking the larvae and by the use of lz-gal4,uas-GFP transgene. Also, the presence of PPO could be analyzed by antibodies to decipher the nature of the defect. Survival of axenic and conventionally raised animal should be added to confirm the model.

---

## [Author Response]

*Essential revisions:*

*Although the experimental set-up and work is highly appreciated, the presented data are in the view of the three reviewers not waterproof to fully support the claim that 1) the intrauterine larval recognition of the obligate Wigglesworthia symbiont regulates the expression of the larval OBP6 and 2) this is an evolutionary conserved component of the insect innate system. The three reviewers raised three important issues, each of them should be addressed before acceptance.*

*1) Specificity*

*The authors claim that the regulation of the larval OBP6 expression and the subsequent production of the crystal cells are regulated by the obligate symbiont Wigglesworthia. This is based upon the sole observation that the deprived obp6 expression in 1st / 2nd instar larvae of aposymbiotic flies (= offspring of tetracycline-treated females that lack all microbiota i.e. Wigglesworthia, Sodalis and Wolbachia present in tsetse lab colonies) is restored by the supplementation of the blood meal given to the mother flies with a Wigglesworthia extract. This 'extract' is actually a homogenate of the bacteriome of WT flies that is a cocktail of the Wigglesworthia bacterium with cellular/molecular components of the tsetse anterior midgut. This means that the presented experimental work is only suggestive for a possible involvement of Wigglesworthia in the larval obp6 expression but is not conclusive. Moreover, it is not clear how Wigglesworthia or its components that are administered through a blood meal could finally reach the larva that is developing in the fly uterus. In previous work, the authors showed that Wigglesworthia cannot colonize aposymbiotic flies when administered this way (Weiss et al.2012). In previous work the authors used an ampicilline-treatment approach to have tsetse flies that produce offspring that is specifically cleared for Wigglesworthia but not for the other microbiota. The use of use this model for the experimental work could make the link between Wigglesworthia/OBP6/crystal cells more conclusive. Also, an additional experimental group where the blood meal is supplemented with the bacteriome of aposymbiotic tsetse (contain tsetse midgut components but devoid of Wigglesworthia) would be at least a more appropriate control group in experiment 1F.*

The reviewers are correct in stating that the extracts of *Wigglesworthia*-containing bacteriome used for supplementation also includes cellular/molecular components of tsetse’s anterior midgut (which presumably also contains *Sodalis* cells). Based on this circumstance, we agree that using bacteriome tissue from offspring of ampicillin treated females (these flies house *Sodalis*, but not *Wigglesworthia*, in their bacteriome/anterior midgut region) is a good control because it accounts for the effects of the cellular/molecular components associated with this tissue. Thus, we performed the experiment again, this time using bacteriome extracts from tsetse reared in the absence of only Wigglesworthia. Also, as the reviewers suggested, we replaced the Gmm^Apo/NB^ control with one that contained extracts of bacteriome from Gmm^Apo^ flies. These changes required that we also modify the designations of some of the treatments, as reflected in the Experimental Procedures, on the x-axis for Figure 1, and in the corresponding figure legend of the revised manuscript.

*2) A better characterization of Tsetse crystal cell*

*The identification of crystal cells in the tsetse fly appears to be based entirely on the number of melanized spots in the cuticle after heat shock. Previous work by different authors used microscopical analysis of crystal cells to develop and validate the reliability of this assay for Drosophila. Microscopical demonstration of crystal cells in untreated tsetse and their depletion in aposymbiotic or RNAi-treated tsetse would give confidence in the reliability of the assay in tsetse.*

The reviewers indicate that our quantification of crystal cells is based exclusively on the number of melanized spots present in heat shocked tsetse larvae. The reviewers suggest using microscopical analysis to further quantify these cells. We have extensively observed/studied cells present in tsetse’s hemolymph under the microscope, but do not feel confident that we can accurately differentiate distinct hemocyte sub-types. Additionally, we previously attempted to detect specific hemocyte subtypes using subtype-specific antibodies (provided by Dr. István Andó). However, these antibodies failed to cross-react with any tsetse hemocytes.

We feel that other results present in the manuscript support our finding that siOBP6 offspring house a reduced population of crystal cells. Specifically, we performed qPCR, using lozenge specific primers, on larval cDNA, and PPO western blots on hemolymph from siOBP6 adults. Our results demonstrated that 1) siOBP6 larvae express significantly fewer lozenge transcripts than do control, and 2) siOBP6 adults produce significantly less PPO than do age-matched wild-type counterparts. Because lozenge expression is required for prohemocytes to differentiate into crystal cells, and because crystal cells are a prominent producer of PPO in *Drosophila* (see Binggeli et al., 2014 and references cited therein), we feel that these results provide evidence that siOBP6 tsetse house a depleted population of these cells.

*3) The part on Drosophila should be better documented*

*The part on Drosophila is really exciting, suggesting a conserved mechanism. Unfortunately, it is too short in the present version, and straightforward experiments should be added to reinforce the conclusion. The impact of microbiota on obp28a and lozenge expression should be repeated in another genetic background. The study will be strongly reinforced by the use of obp28a mutation (a temporally feasible with the advent of CRISPR), or at least the use of another independent RNAi to confirm the first one. In which tissues is obp28a expressed in larvae? The status of crystal cells could be tested by cooking the larvae and by the use of lz-gal4,uas-GFP transgene. Also, the presence of PPO could be analyzed by antibodies to decipher the nature of the defect. Survival of axenic and conventionally raised animal should be added to confirm the model.*

We too are very excited about demonstrating that *Drosophila*’s microbiota regulates a conserved hematopoietic pathway in their host. However, we want to respectfully emphasize to the reviewers that the priority of this paper lies with the tsetse fly, and that the *Drosophila* component is an intriguing one that we plan to investigate in more detail in future studies. With this in mind, we have performed the following additional experiments that we feel are necessary to demonstrate a positive correlation between *Drosophila*’s microbiota, *obp28a* expression and the production of crystal cells (and thus PPO and melanin). The results of these new experiments solidify data presented in the original draft of the manuscript. Specifically, we:

1) We quantified *obp28a* and lozenge expression in genetically distinct Oregon^R^ and w^118^
*Drosophila* lines, both axenically and conventionally reared (Figure 5).

2) We used Oregon^R^ and w^118^
*Drosophila* lines to show that axenic larvae house conspicuously fewer crystal cells than do conventionally reared individuals (Figure 5, top panels).

3) We used Oregon^R^ and w^118^
*Drosophila* lines to show that axenic adults produce less prophenoloxidase than do conventionally reared individuals (Figure 5, bottom panels).

4) We performed survival experiments using a *Drosophila obp28a* deletion mutants (CRISPR generated). These experiments confirmed that flies expressing reduced quantities of this gene are more susceptible to clean wounds than are their wild-type counterparts (Figure 5). Furthermore, *obp28a* deletion mutants fail to generate a melanotic scab at the wound site while wild-type flies do (Figure 5).